# Dysregulated UPR and ER Stress Related to a Mutation in the *Sdf2l1* Gene Are Involved in the Pathophysiology of Diet-Induced Diabetes in the Cohen Diabetic Rat

**DOI:** 10.3390/ijms24021355

**Published:** 2023-01-10

**Authors:** Chana Yagil, Ronen Varadi-Levi, Chen Ifrach, Yoram Yagil

**Affiliations:** 1Laboratory for Molecular Medicine and Israeli Rat Genome Center, Barzilai University Medical Center, Ashkelon 7830604, Israel; 2Faculty of Health Sciences, Ben-Gurion University of the Negev, P.O. Box 653, Beer Sheva 8410501, Israel

**Keywords:** type 2 diabetes, glucose, insulin, rodent model, diet, mechanism, gene mutation, protein expression, unfolded protein response, endoplasmic reticulum stress

## Abstract

The Cohen Diabetic rat is a model of type 2 diabetes mellitus that consists of the susceptible (CDs/y) and resistant (CDr/y) strains. Diabetes develops in CDs/y provided diabetogenic diet (DD) but not when fed regular diet (RD) nor in CDr/y given either diet. We recently identified in CDs/y a deletion in *Sdf2l1*, a gene that has been attributed a role in the unfolded protein response (UPR) and in the prevention of endoplasmic reticulum (ER) stress. We hypothesized that this deletion prevents expression of SDF2L1 and contributes to the pathophysiology of diabetes in CDs/y by impairing UPR, enhancing ER stress, and preventing CDs/y from secreting sufficient insulin upon demand. We studied SDF2L1 expression in CDs/y and CDr/y. We evaluated UPR by examining expression of key proteins involved in both strains fed either RD or DD. We assessed the ability of all groups of animals to secrete insulin during an oral glucose tolerance test (OGTT) over 4 weeks, and after overnight feeding (postprandial) over 4 months. We found that SDF2L1 was expressed in CDr/y but not in CDs/y. The pattern of expression of proteins involved in UPR, namely the PERK (EIF2α, ATF4 and CHOP) and IRE1 (XBP-1) pathways, was different in CDs/y DD from all other groups, with consistently lower levels of expression at 4 weeks after initiation of DD and coinciding with the development of diabetes. In CDs/y RD, insulin secretion was mildly impaired, whereas in CDs/y DD, the ability to secrete insulin decreased over time, leading to the development of the diabetic phenotype. We conclude that in CDs/y DD, UPR participating proteins were dysregulated and under-expressed at the time point when the diabetic phenotype became overt. In parallel, insulin secretion in CDs/y DD became markedly impaired. Our findings suggest that under conditions of metabolic load with DD and increased demand for insulin secretion, the lack of SDF2L1 expression in CDs/y is associated with UPR dysregulation and ER stress which, combined with oxidative stress previously attributed to the concurrent *Ndufa4* mutation, are highly likely to contribute to the pathophysiology of diabetes in this model.

## 1. Introduction

Diabetes mellitus in humans is likely to be a heterogeneous disease, with different complex mechanisms predominating in the various subtypes of diabetes [1,2,3]. Dissection of the discrete mechanisms involved is challenging in humans and more amenable in animal models of the disease [4,5].

The Cohen Diabetic rat represents a genetically inbred non-obese model of diabetes, which phenotype is induced following exposure to a custom-prepared diabetogenic diet (DD) and is, therefore, nutrition dependent [6,7]. The model consists of the inbred Cohen Diabetes genetically susceptible (CDs/y) and resistant (CDr/y) strains. Susceptibility to diabetes is based on the metabolic phenotype following exposure to DD. CDs/y provided standard rat chow (RD) remains normoglycemic but when exposed to DD, invariably develops within 4 weeks a diabetic phenotype that is reminiscent of type 2 diabetes in humans. CDr/y, on the other hand, remains non-diabetic throughout its lifetime, irrespective of the type of diet. The dietary intervention in this model thus constitutes an important and dominant environmental factor, which allegedly imposes a metabolic strain that leads to the development of diabetes in the sensitive CDs/y but not in the resistant CDr/y rat strain.

In the investigation of the pathophysiology of diabetes in the Cohen rat model, we previously identified in CDs/y through linkage analysis a quantitative trait locus (QTL) on rat chromosome 4 that incorporates NADH dehydrogenase (ubiquinone) 1 alpha sub-complex 4 (*Ndufa4*), a nuclear gene that affects mitochondrial function [8]. We subsequently detected in CDs/y through next generation sequencing a major deletion in that gene that prevents expression of the NDUFA4 protein [9]. We demonstrated in CDs/y provided DD that in the absence of NDUFA4, mitochondrial complex I activity is reduced, resulting in oxidative stress [10]. We thereby established an association between the *Ndufa4* mutation in CDs/y, mitochondrial dysfunction, and oxidative stress. This triad, which had been previously implicated in impairing the ability of the pancreas to secrete insulin in other animal models of diabetes, as well as in diabetic humans [11], did not provide, a sufficiently comprehensive explanation for the full development of the diabetic phenotype in the Cohen susceptible CDs/y strain.

In additional studies in which we performed whole genome sequencing and genome-wide screening, we identified in CDs/y a second genomic locus of interest on rat chromosome 11 that we had not previously detected by linkage analysis [9,10]. This locus consists of a high-impact 907 bp deletion that coincides with the *Sdf2l1* (Stromal cell-derived factor 2 like 1) gene (NM_001109433, chr11: 88,122,327–88,123,234) and that involves two out of three exons in that gene. The deletion was unique to CDs/y among 24 other studied rodent strains [9]. Under normal conditions, the translational product of the gene, the SDF2L1 protein, is involved in activation of the mammalian unfolded protein response (UPR), a cellular response to increased demand for protein synthesis that prevents excess accumulation of unfolded proteins in the endoplasmic reticulum (ER) and helps minimize or altogether avoid ER stress [12,13].

The molecular pathways underlying UPR are complex. As shown schematically in Figure 1, UPR is initiated through 3 ER-resident membrane-bound proteins, protein kinase RNA-like ER kinase (PERK), inositol requiring enzyme 1 (IRE1) and activating transcription factor 6 (ATF6), each triggering a distinct downstream signaling pathway [14]. Under baseline conditions, PERK, IRE1 and ATF6 are coupled to a binding immunoglobulin protein (BIP/GRP78) that holds them in an inactive state on the ER membrane. During increased demand for protein synthesis (such as glucose loading requiring increased insulin secretion), unfolded proteins tend to accumulate in the ER, resulting in ER stress. In response, BiP is released from the membrane proteins and the PERK, IRE1 and ATF6 pathways are triggered, initiating the unfolded protein response [15]. The release of BiP from the membrane-bound proteins is enabled by associating with three additional ER-resident proteins, ERdj3, SDF2 (constitutively expressed) and SDF2L1 (expression induced during ER stress), the resulting complex inhibiting further aggregation and accumulation of misfolded proteins [12,16,17,18,19]. As the triggered UPR prevents additional protein synthesis [15,20,21,22], the two processes together avert further ER stress.

As ER stress has previously been implicated in the development of diabetes in other models of the disease [16,22,23,24], the aim of our current study was to investigate the hypothesis that the *Sdf2l1* mutation we previously found in CDs/y prevents expression of SDF2L1, precluding thereby BIP from being effectively recruited as a chaperone and hindering the triggering of the PRK, IRE1 and ATF6 pathways. As result, UPR is impaired, leading to the development of ER stress, the inability to secrete insulin in sufficient amounts during metabolic loading (such as feeding with DD) and resulting in the evolution of the diabetic phenotype.

## 2. Results

### 2.1. SDF2L1 Protein Expression

In tissue homogenates from the pancreas, liver, and kidneys, SDF2L1 was clearly expressed in CDr/y but not in CDs/y, as shown in Figure 2.

### 2.2. Expression of Proteins Involved in UPR

Data are provided in animals provided RD or DD from day 4 of the study and at weekly intervals thereafter over 4 weeks (by which time the diabetic phenotype becomes overt in CDs/y DD but not in any of the other groups. We analyzed the data (1) for *group comparison* by comparing the expression of the protein in each group (CDr/y DD, CDs/y RD and CDs/y DD) to that in CDr/y RD at each time point (from day 4 day to 4 weeks), and (2) for *change in expression* by studying the change in protein expression within group over time (from day 4 to 4 weeks).

**GRP78/BiP** The data are shown in Figure 3, panels A–C. *Group comparison* showed that expression level in CDr/y DD (Figure 3A) was similar to CDr/y RD at all time points, except at 4 weeks when it was significantly lower; in CDs/y RD (Figure 3B), expression level was lower than in CDr/y RD at 2 weeks but not different at all other time points; in CDs/y DD (Figure 3C), there was no difference in level of expression compared to CDs/r RD at any time points. *Change in expression* analysis showed that in CDr/y DD (Figure 3A), there were no significant changes in expression during the 4 weeks of the study (F = 1.9382, *p* = n.s.); in CDs/y RD (Figure 3B), expression level increased significantly at 3 weeks (*p* = 0.0044 versus 4 days), but returned to baseline at 4 weeks; in CDs/y DD (Figure 3C), no significant change in expression was detected throughout the 4 weeks of the study (F = 1.7450, *p* = n.s.).

### 2.3. PERK Pathway Proteins

**Phospho eif/Eif2α** The data are shown in Figure 3, panels D-F. *Group comparison* showed that the ratio of expression of Phospho eif/Eif2α in CDr/y DD (Figure 3D) was lower than in CDr/y RD at 1 and 2 weeks but not at 4 weeks; in CDs/y RD (Figure 3E), the ratio was lower than in CDr/y RD at 1 week but not different at all other time points; in CDs/y DD (Figure 3F), the ratio was significantly lower than in CDr/y RD at 1, 2 and 4 weeks. *Change in the ratio of expression* analysis showed that in CDr/y DD (Figure 3D), the ratio dropped after 1 and 2 weeks (*p* = 0.0000 and *p* = 0.0010, respectively) but increased above baseline at 4 weeks (*p* = 0.0268); in CDs/y RD (Figure 3E), the ratio decreased significantly at 1 and 2 weeks (*p* = 0.0008 and *p* = 0.0177, respectively) but returned to baseline at 4 weeks; in CDs/y DD (Figure 3F), there was no significant change in the ratio of expression over time (F = 1.5251, *p* = n.s.).

**ATF4** The data are shown in Figure 3, panels G-I. *Group comparison* showed that expression level in CDr/y DD (Figure 3G) compared to CDr/y RD was lower at 4 days and 1 week but not different at 2, 3 and 4 weeks; in CDs/y RD (Figure 3H), compared to CDr/y RD, expression level was significantly lower at 1 week, not different at 2 weeks, increased at 3 weeks and not different again at 4 weeks; in CDs/y DD (Figure 3I), expression level compared to CDr/y RD was significantly lower at 1, 3 and 4 weeks. *Change in expression* analysis showed that in CDr/y DD (Figure 3G), expression level of ATF4 compared to day 4 increased significantly after 2 and 3 weeks (*p* = 0.0000 and *p* = 0.0010, respectively) but only tended to increase at 4 weeks (*p* = 0.0991); in CDs/y RD (Figure 3H), the expression level increased significantly at 2, 3 and 4 weeks (*p* = 0.00128, *p* = 0.0000 and *p* = 0.0003, respectively); in CDs/y DD (Figure 3I), the expression level decreased or tended to decrease at 1, 3 and 4 weeks (*p* = 0.0452, *p* = 0.0587 and *p* = 0.0000, respectively).

**CHOP** The data are shown in Figure 3, panels J-L. *Group comparison* showed that expression level in CDr/y DD (Figure 3J) compared to CDr/y RD was lower at 4 days and at 1 and 2 weeks but was not different at 3 and 4 weeks; in CDs/y RD (Figure 3K), when compared to CDr/y RD, expression was significantly lower at 4 days and 1 week, but not different at the other time points; in CDs/y DD (Figure 3L), expression compared to CDr/y RD was not different at 4 days, lower at 1 week, not different at 2 and 3 weeks and significantly lower again 4 weeks. *Change in expression* analysis showed that in CDr/y DD (Figure 3), expression level of ATF4 did not significantly change over 4 weeks (F = 0.0041, *p* = n.s.); in CDs/y RD (Figure 3K), the expression level did not significantly change over time either (F = 1.778, *p* = n.s.); in CDs/y DD (Figure 3L), the expression level decreased significantly at 1 and 4 weeks (*p* = 0.0102 and *p* = 0.0000, respectively).

**IRE 1 pathway protein** Expression of XBP was studied at 1 and 4 weeks only. Data are shown in Figure 3, panels M-O.

**XBP1***Group comparison* showed that expression level in CDr/y DD (Figure 3M) compared to CDr/y RD was significantly lower at week 1 but not at week 4; in CDs/y RD (Figure 3N), when compared to CDr/y RD, expression was significantly lower at week 1 but not at week 4; in CDs/y DD (Figure 3O), expression compared to CDr/y RD was significantly lower at 1 and 4 weeks. *Change in expression* analysis showed that compared to 1 week, the level of expression in CDr/y DD (Figure 3M) did not significantly change after 4 weeks (F = 6.9586, *p* = n.s.); in CDs/y RD (Figure 3N), expression did not significantly change over time either (F = 0.749, *p* = n.s.); in CDs/y DD (Figure 3O), expression did not change (F = 1.6510, *p* = n.s.).

### 2.4. Glucose Tolerance and Insulin Secretion

*OGTT* The results are shown in Figure 4.

*Baseline OGTT* (prior to dietary intervention): At time 0, there was no difference between CDs/y and CDr/y fed RD neither in blood glucose levels (Figure 4A) nor in insulin/glucose ratio (Figure 4C) (*p* = 0.40 and *p* = 0.02, respectively). In the “early phase” of the OGTT at 15 min, glucose levels already tended to be higher in CDs/y DD than in CDs/y RD (*p* = 0.06). At 30 min, glucose levels were significantly higher in CDs/y than in CDr/y (*p* < 0.01), whereas the insulin/ glucose ratio was higher in CDr/y than in CDs/y (*p* < 0.01). In the latter phases of the OGTT, glucose and insulin/glucose were no longer different between the sub-strains at 60 min (*p* = 0.86, *p* = 0.334, respectively) and 180 min (0.49, 0.24, respectively).

*OGTT after 4 weeks of feeding with DD*: At time 0, glucose levels were similar in CDs/y DD and CDr/y DD (Figure 4B), and yet insulin/glucose values were already higher in CDr/y (Figure 4D). At all other time points, glucose levels were significantly higher in CDs/y than in CDr/y (Figure 4B), whereas insulin/glucose values were significantly higher in CDr/y than in CDs/y (Figure 4D).

*Postprandial:* The results of postprandial plasma glucose and insulin levels are shown in Figure 5A–D, respectively. One week from initiation of the dietary intervention, post-prandial glucose levels (PPGL) were already slightly, yet significantly, higher in CDs/y than in CDr/y irrespective of diet, whereas the post-prandial insulin/glucose ratio (PPIGr) was lower in CDs/y DD than in any of the other groups. As of 2 weeks, PPGL gradually increased in CDs/y DD to levels significantly above those in the other three groups, achieving the highest levels at 4 months. At 2 weeks, PPIGr increased in CDr/y DD compared to CDr/y RD; in CDs/y, on the other hand, PPIGr was significantly lower than in CDr/y. This pattern persisted thereafter, the PPIG dropping to a nadir in CDs/y DD at 2.5 and 4 months.

*Insulin content:* The amount of insulin in the pancreas of CDs/y RD, as shown in Figure 6, was not different from that found in CDr/y RD. After 4 weeks of DD, the amount of insulin increased significantly and equally in CDs/y and CDr/y, demonstrating that CDs/y pancreas is producing insulin such as CDr/y when exposed to DD, even though plasma insulin is lower in CDs/y.

## 3. Discussion

Our previous investigation of the pathophysiology of diabetes in the Cohen rat model using linkage analysis, construction of congenic and consomic strains and whole genome sequencing led to findings that explained in part the diabetic phenotype of CDs/y [7,8,9,10]. In the current study, we pursued the exploration of the pathophysiology of the disease in our model by focusing on the functional significance of the *Sdf2l1* gene mutation, which we had recently detected in the diabetes-prone CDs/y strain [9,10].

In our investigation, we first established and successfully confirmed that the *Sdf2l1* mutation in the diabetes prone CDs/y strain prevents altogether translation of the gene into SDF2L1. We then set out to determine the functional significance of the absence of the SDF2L1 protein in CDs/y and whether it was likely to be involved in the pathophysiology of diabetes in our model. Since SDF2L1 has been attributed a central role in the initiation of UPR [12,22,25], we tested the hypothesis that the absence of SDF2l1 would impair UPR and, thereby lead to sustained ER stress in CDs/y.

We opted to evaluate UPR, a challenging task, by measuring the level of expression of several of the key proteins involved [26,27,28] over 4 weeks of RD or DD, the time required for the diabetic phenotype to evolve in CDs/y DD. During evaluation of the level of expression of GRP78/Bip in CDs/y, one of the key players in triggering UPR, we did not detect any significant changes over time in expression levels compared to CDr/y RD. This finding is consistent with our hypothesis that in the absence of SDF2L1, mobilization of this protein is prevented and initiation of UPR, which might have prevented ER stress, does not occur. In evaluating UPR downstream along the PERK pathway, we observed that the pattern of expression over time of the proteins EIF2α, ATF4 and CHOP was distinctly different in CDs/y DD from that in all the other groups at 4 weeks, at which time the diabetic phenotype evolves in full in CDs/y DD but not in the other groups. In CDs/y DD, the level of expression was consistently and significantly lower from that in CDr/y RD, suggesting that at the time point CDs/y became diabetic, the proteins participating in UPR were consistently under-expressed in this strain. In contrast, in CDr/y DD and CDs/y RD, expression of these three proteins at 4 weeks was similar to CDr/y RD. While studying the expression of the IRE1 pathway protein XBP-1 [29], we found a similar pattern, i.e., CDr/y DD and CDs/y RD had a similar level of expression as CDr/y RD at 4 weeks, while in CDs/y DD, the level of expression remained significantly lower at that time point. Taken together, the lower levels of expression in CDs/y DD of key proteins participating in both PERK and IRE pathways at 4 weeks of feeding with DD can be interpreted as dysregulation, or impairment of the UPR response, which renders the animal prone to ER stress. 

Can the impaired UPR response in face of the 4-week metabolic load in CDs/y due to the absence of expression of SDF2L1 be targeted and modulated? Is there a way of bypassing the SDF2L1-dependent release of BiP/GRP78 from any of the three ER-resident proteins (PERK, IRE1, ATF6), a step required for their respective activation? There appears to be at least one possibility of activating the PERK pathway, downstream of PERK, using salubrinal, a known ER stress modulator that has been shown to inhibit dephosphorylation of eIF2~P and that has been successfully used as such in experimental models [30,31,32]. As the activity of eIF2 is in its phosphorylated form, salubrinal would increase the levels of eIF2~P, resulting in downstream activation of the PERK pathway of the unfolded protein response (UPR) independently of PERK activation. Extrapolating to our diabetic rodent model that is SDF2L1 deficient, salubrinal could become an interesting modality to modulate and improve UPR and reverse, at least in part, the diabetic phenotype that we attribute to dysregulated UPR. Although we would be targeting only one out of the three UPR pathways and we cannot ascertain the relative importance or contribution of each of the pathways to averting ER stress, this is a direction worth investigating.

If UPR is indeed dysregulated in CDs/y after the animals are fed with DD for 4 weeks, could this lead to impaired ability of CDs/y to secrete insulin under conditions of high demand, such as during acute glucose loading? We set out to answer this question by investigating the ability of CDs/y provided DD to secrete insulin during an oral glucose tolerance test, with the response of CDr/y serving as reference and with CDs/y provided RD serving as control for the effect of DD. We found that during the baseline OGTT, which we performed prior to the dietary intervention with DD, CDs/y RD have in the early phase of the OGTT (30 min) a reduced ability to secrete insulin compared to CDr/y RD, whereas in the late phase (60–180 min), the insulin response was normal. This finding in the early phase of the OGTT [33], which reflects release of “prefabricated” insulin from storage vesicles, is consistent with impaired secretion but not with impaired production of insulin. The impairment appears to be mild and transient, as in the latter late phase of the OGTT, glucose levels and the insulin/glucose ratio normalized. Such abnormality in the insulin response to glucose [34] has previously been suggested to precede the development of type 2 diabetes [35]. These findings suggest that under normal conditions and despite the *Sdf2l1* mutation, CDs/y fed RD maintain the ability to secrete insulin into the circulation in amounts that are below normal, and yet sufficient to maintain normoglycemia. In contrast, while studying the effect of repeated daily administration of DD over 4 weeks on the ability of CDs/y to secrete insulin in response to acute glucose loading, glucose levels in CDs/y were significantly higher than in CDr/y at all time points during the OGTT, except for time 0, whereas the ability of CDs/y to secrete insulin in response to the glucose load was consistently below that of CDr/y. These finding suggest that the dietary intervention (DD) exerts a more profound and lasting detrimental effect on the ability of CDs/y to secrete insulin, beyond the underlying inability to respond in the early phase of the insulin secretion and affecting the late phase insulin response, eventually leading to the development of the diabetic phenotype.

Studying postprandial glucose and insulin/glucose ratio over 4 months was further informative as to the evolution of the diabetic phenotype in CDs/y. As of 2 weeks after initiating DD, the significantly higher postprandial glucose levels were indicative of early impaired glucose handling in CDs/y DD (earlier than previous timing of the 4 week requirement for the development of the diabetic phenotype). The magnitude of hyperglycemia and reduced insulin secretion gradually increased over 4 months, thereby defining more accurately and extensively the course of the diabetic phenotype in this genetically inbred animal strain. Interestingly, the lower postprandial insulin/glucose ratio in CDs/y irrespective of diet as of 1 week into the study adds further support the conclusion that CDs/y has an underlying inability to secrete sufficient insulin in response to dietary load when compared to CDr/y. Despite the lack of consistent differences in insulin/glucose ratio between CDs/y DD and CDs/y RD during the first 4 weeks of the study, glucose levels in CDs/y DD were already higher than in CDs/y RD or CDr/y on either diet as of 2 weeks. At 2.5 and 4 months of the study, glucose levels were significantly higher whereas insulin/glucose ratio was lower in CDs/y DD compared to all other groups, consistent with the continuing evolution of diabetic phenotype over time, consistent with the diabetic phenotype in this strain.

Is the inability of CDs/y to secrete sufficient insulin due to an impairment in insulin production (synthesis), secretion, or both? The results of Western blotting of insulin in the pancreas in CDs/y and CDr/y indicate that when both strains are fed RD, they can produce (synthesize) similar amounts of insulin. More important, however, is our demonstration that both strains also maintain the ability to produce in the pancreas equal amounts of insulin when provided DD over 4 weeks, despite the development of the diabetic phenotype. These latter findings suggest that in both CDs/y DD and CDr/y DD, insulin is produced and stored within the pancreas, yet in CDs/y DD, the amount of insulin that is secreted out of the pancreas and into the circulation is insufficient and disproportionally low in proportion to glucose levels, as result of impaired ability of CDs/y to secrete insulin out of the insulin-producing cell [11].

## 4. Methods

### 4.1. Animals

We procured the animals from the Cohen Diabetic susceptible and resistant colonies, inbred and maintained at the Israeli Rat Genome Center (IRGC) of the Barzilai University Medical Center in Ashkelon, Israel [6,7]. We housed the animals in accordance with institutional and governmental regulations, principles of laboratory animal care (NIH publication no. 85-23, revised 1985) and guidelines of the American Society of Physiology for the Care and Use of Laboratory Animals. We maintained climate-controlled conditions and adjusted light to regular timed diurnal cycles. The study protocol was reviewed and approved by the Institutional Animal Care and Use Committee (Animal Experimentation Committee).

Unless stated otherwise, experiments included four groups of animals in a 2 × 2 design: CDs/y provided DD (experimental group), or RD (diet control) and CDr/y fed DD (strain control) or RD (strain and diet control group). After weaning, we provided the animals with standard rat chow (RD) and tap water ad libitum. When required for experimental purposes, we switched the diet in the relevant groups at age 6–7 weeks to custom-prepared DD, adding copper-poor water for drinking ad libitum, as previously described [6]. After 4 weeks on DD, CDs/y but not CDr/y, invariably develop a diabetic phenotype, which consists of an abnormally exaggerated hyperglycemic response to glucose loading and impaired insulin secretion [6,7].

### 4.2. Tissue Extraction

To extract tissue (pancreas, kidney, liver) for protein analysis, we anesthetized the animals by intraperitoneal injection of a mixture of xylazine (10 mg/kg)/ketamine (100 mg/kg), 0.15 mL/100 g body weight, sacrificed them by exsanguination through the bifurcation of the aorta and surgically excised the targeted tissues. We snap-froze the tissues in liquid nitrogen and stored them at −80 °C until processed further.

### 4.3. Protein Extraction

We washed 50–100 μg frozen tissue and minced in RIPA-PI (protease inhibitor cocktail from Sigma-Aldrich, St. Louis, MO, USA), dounce homogenized on ice and centrifuged the homogenate at 4000 rpm (1500× *g*), obtaining a membrane-rich pellet and the supernatant consisting of the cytosolic fraction. We used the Bradford method to quantitate the amount of protein extracted.

### 4.4. Western Blotting

To demonstrate protein expression in excised tissues, we loaded 50–70 ug protein samples on Any kD denatured gels (Bio-Rad), ran the gels, and transferred the samples to PVD membrane. After blocking the membrane with 5% milk in PBST (0.1%) or 5% bovine albumin (BSA) for 1hr, we incubated with 1st antibody overnight at 4 °C, using the appropriate antibody for each protein. We then washed the membrane with PBST 0.1% Tween and incubated for 1 h with the 2nd antibody. We developed the membranes with Clarity Western ECL Substrate (Bio-Rad, Hercules, CA, USA). As reference protein, we used tubulin (Sigma T5168). We analyzed the gels by densitometry using the web-based Gel Analyzer software (http://www.gelanalyzer.com), correcting each sample for tubulin.

In order to overcome between-gel variation, we added to each gel the same five duplicate samples from CDr/y-RD (aged 8–9 weeks), which served as reference standard. We expressed the results as percent of the average CDr/y-RD standard reading on the same gel at each time point of the study, allowing thereby between-gel comparison.

### 4.5. SDF2L1 Expression

To determine the effect of the *Sdf2l1* gene mutation found in CDs/y on its expression at the protein level, we studied tissue expression of the SDF2L1 protein by Western blot using the SDF2L1 Abcam antibody (1:200) in homogenates of the pancreas, liver, and the kidney. As positive control, we studied the protein expression in CDr/y, in which the *Sdf2l1* mutation had not been detected.

### 4.6. Expression of Proteins Involved in UPR

To evaluate the unfolded protein response (UPR) in CDs/y and CDr/y provided RD or DD, we studied protein expression in homogenates of the pancreas in CDs/y and CDr/y after 4 days, 1 week, 2 weeks, 3 weeks, and 4 weeks from initiation of DD or RD. We measured by Western blot the expression level of GRP78/BiP and of key proteins involved in initiating UPR: the PERK (eukaryotic initiation factor 2α -EIF2α, Activating Transcription Factor 4-ATF4 and the C/EBP homologous protein–CHOP) and the IRE (X box binding protein 1-XBP-1) pathways. As shown in Figure 1, PERK activation induces EIF2α phosphorylation which increases transcription of AFT4 that increases in turn CHOP levels and IRE1α activation increases transcription of XBP-1 (and JNK which we did not study). In order to study expression of GRP78/BiP, we used as primary antibody Anti-GRP78 from Abcam ab (1:50,000) (75kDa). For Phospho eif2α/Eif2α, we used Anti-elF2α (38 kDa, 1:1000) and Anti-Phospho-elF2α (38kDa, 1:1000) from Cell Signaling technology. For ATF-4, we used Anti-ATF-4 (EPR18111) from Abcam ab184909 (50 kDa, 1:700). For CHOP, we used Anti DDIT3 (Chop) from Abcam ab179823 (25 kDa, 1:1000). For XBP1, we used Anti XBP-1S (E8Y5F) from Cell Signaling technology (50 kDa, 1:1000).

### 4.7. Ability of CDs/y to Secrete Insulin

Since impaired regulation of UPR is expected to affect the ability to increase insulin secretion during high demand (RD–low demand, DD–high demand), we investigated insulin secretion in CDs/y during feeding with RD and in response to DD. We evaluated the insulin response to dietary loading during an acute oral glucose tolerance test (OGTT) and after nocturnal feeding (post-prandial) with DD. We used the insulin response of CDr/y as the normal control.

*Oral glucose tolerance test (OGTT)* We performed an OGTT in CDs/y and CDr/y at two weeks after weaning from maternal feeding and while the animals were fed RD. We repeated the OGTT after feeding both sub-strains for 4 weeks with either DD or RD. We performed the OGTT as previously described [6,7], sampling blood from the tip of the tail for glucose at times 0, 15, 30, 60, 120 and 180 min and for insulin at times 0, 30, 60 and 180 min.

*Post-prandial test (PPT)* We measured postprandial glucose and insulin levels in CDs/y and CDr/y fed either RD or DD points after 1, 2, 3 and 4 weeks, 2.5 months, and 4 months. Since rodents normally feed overnight, we sampled blood from the tip of the tail between 07:00 and 08:00 a.m.

In order to measure glucose levels, we used standard glucometers. To measure insulin levels, we used the Mercodia insulin assay (Rat Insulin ELISA) from Mercodia, Uppsala, Sweden.

### 4.8. Insulin Content in the Pancreas

If in the absence of SDF2L1, the ability of CDs/y to secrete insulin in response to glucose loading were impaired, we investigated whether this would be due to reduced insulin production (synthesis), normal production but impaired secretion from the islets of Langerhans to the peripheral blood, or both. As a measure of the ability of CDs/y to produce insulin, we evaluated insulin content in the pancreas of CDs/y by Western blot, prior to and after feeding the animals with DD for 4 weeks, using CDr/y as reference. We used as insulin antibody the mouse mAb #8138 which is cross-reactive with the rat (6 kDa, 1:1000) from Cell Signaling Technology, Danvers, MA, USA.

### 4.9. Statistics

Data are provided as mean +/− standard error of mean. For between group analysis, we used either the Student’s *t*-test (two tailed) or one-way analysis of variance (ANOVA), and the Least Significance Difference (LSD) testing for post-hoc analysis, as applicable. We set significance level at *p* < 0.05. For statistical analyses, we used the Statistica Software version 14.0.

## 5. Conclusions

We conclude that the evidence gathered in the current study indicates that in CDs/y, SDF2L1 is not expressed, UPR is dysregulated after 4 weeks of DD, insulin secretion but not production is impaired and that the impairment in insulin secretion is aggravated following short and long-term feeding with DD.

How do we link these findings and do they further our understanding of the pathophysiology of diabetes in the Cohen Diabetic rat model of diet induced diabetes? Key to integrating our findings is the realization that there are at least three central players that we can associate with the development of diabetes in our model of complex disease. The first is the dietary intervention with DD, which comprises of a high metabolic load that is deficient in copper. The second is the *Ndufa4* mutation. The third is the *Sdf2l1* deletion. Based on our findings, we propose that the development of diabetes during exposure to DD in CDs/y is induced by multiple “hits”, as most complex traits are, and that we appear to have detected at least two of the hits. One of the hits is allegedly induced by the *Ndufa4* mutation which adversely affects mitochondria complex I function that leads in turn to increased oxidative stress. Oxidative stress is further aggravated by the inability of CDs/y to generate an adequate anti-oxidative response with SOD1 and SOD2 and by the low copper content in diet that also adversely affects complex IV function. The result is a reduction in mitochondrial production of ATP in CDs/y. A second hit is related to the *Sdf2l1* mutation, which adversely affects UPR and leads to ER stress. Together, oxidative and endoplasmic reticulum stress adversely affect the function of the pancreas and impair the ability of pancreatic insulin release in response to glucose loading, leading to the development of diabetes. The novelty in our findings is that, to the best of our knowledge, this is the first time that a direct link has been established in an animal model between genetically transmitted mutations in at least two genes that are associated with mitochondrial dysfunction, oxidative stress and ER stress and the pathophysiology of diet-induced diabetes, as shown in Figure 7.

A limitation of our investigation is that we studied UPR in homogenates of the whole pancreas, as opposed to studies in isolated islets of Langerhans, which might have been more specific. Studies in isolated islets of Langerhans have, however, their own severe limitations, as protein expression might be affected by the islet isolation procedures per se, as well as by the absence of the surrounding natural environment in which the beta cells within the islets normally function and secrete insulin. A further limitation of our study is that in evaluating the unfolded protein response, we had to resort to an indirect method by studying expression of major proteins involved, as opposed to using a highly specific marker of UPR that is currently unavailable.

## Figures and Tables

**Figure 1 ijms-24-01355-f001:**
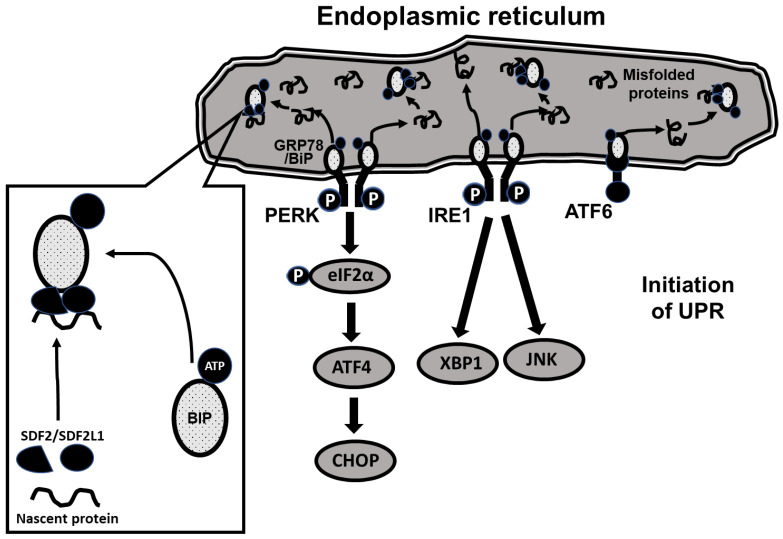
Scheme of the unfolded protein response (UPR) in the endoplasmic reticulum (ER). UPR is initiated through 3 ER-resident membrane-bound proteins, protein kinase RNA-like ER kinase (PERK), inositol requiring enzyme 1 (IRE1) and activating transcription factor 6 (ATF6). Under baseline conditions, PERK, IRE1 and ATF6 are coupled to a binding immunoglobulin protein (BIP/GRP78) that holds them in an inactive state on the ER membrane. During increased protein synthesis misfolded proteins accumulate in the ER. In response, BiP is released from the membrane proteins and the PERK, IRE1 and ATF6 pathways are triggered, initiating the unfolded protein response. The release of BiP from the membrane-bound proteins is enabled by associating with three additional ER-resident proteins, ERdj3, SDF2 (constitutively expressed) and SDF2L1 (expression induced during ER stress), the resulting complex inhibiting further aggregation and accumulation of misfolded proteins. The key proteins involved in the activation of the PERK pathway are the eukaryotic initiation factor 2α (eIF2α), followed downstream by the activating transcription factor 4 (ATF4) and the C/EBP homologous protein (CHOP) and of the IRE1 pathway the X box binding protein 1 (XBP-1) and the c-Jun N-terminal kinase protein (JNK).

**Figure 2 ijms-24-01355-f002:**
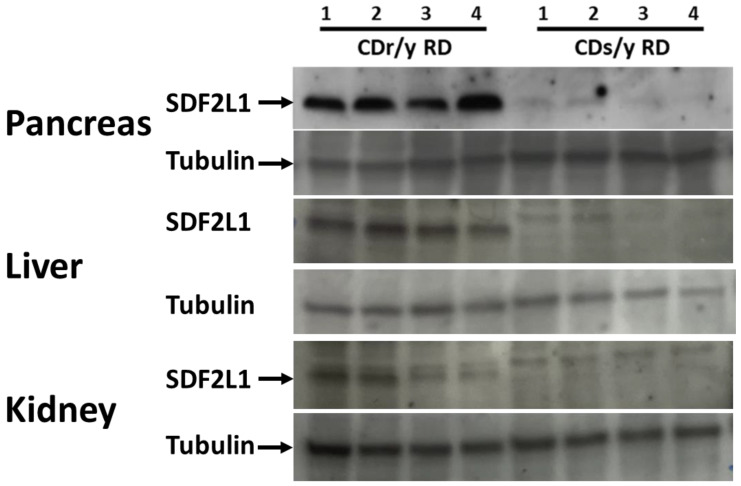
Western blot expression of SDF2L1 (24 kDa) in homogenates of the pancreas, liver, and kidney of CDs/y to CDr/y provided RD. Four separate animals, labeled 1–4, were used for each strain. The reference protein was tubulin (~50 kDa).

**Figure 3 ijms-24-01355-f003:**
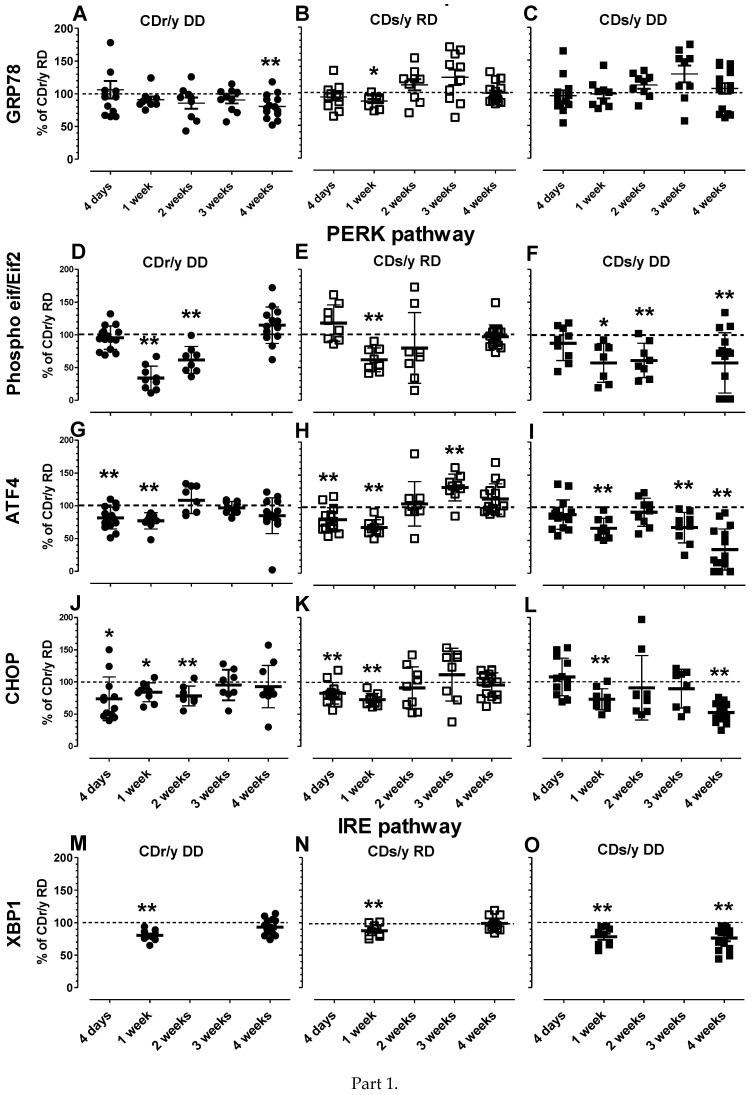
(**Part 1**) Expression of GRP78/BIP (panels **A**–**C**, *n* = 9–30), eIF2α (panels **D**–**F**, *n* = 7–23), ATF4 (panels **G**–**I**, *n* = 9–25), CHOP (panels **J**–**L**, *n* = 8–25) and XBP1 (panels **M**–**O**, *n* = 9–25) in the pancreas shown as scattergrams in CDr/y DD (black filled circles), CDs/y RD (open rectangles), and CDs/y DD (black filled rectangles) respectively, and expressed as percent of expression of CDr/y RD, at 4 days and at weekly intervals thereafter, until 4 weeks of the study. All data were corrected for the same 10 CDr/y-RD pooled in five duplicates, which served as reference in each blot to correct for between variations. The horizontal dotted line represents 100% of mean expression in CDr/y RD. * *p* < 0.05, ** *p* < 0.01 indicates statistical significance by Student’s t-test of values compared to CDr/y at each given time point. (**Part 2**) Representative blots of each of the respective proteins quantified by densitometry in part 1 of the Figure, with alpha-tubulin serving as reference protein.

**Figure 4 ijms-24-01355-f004:**
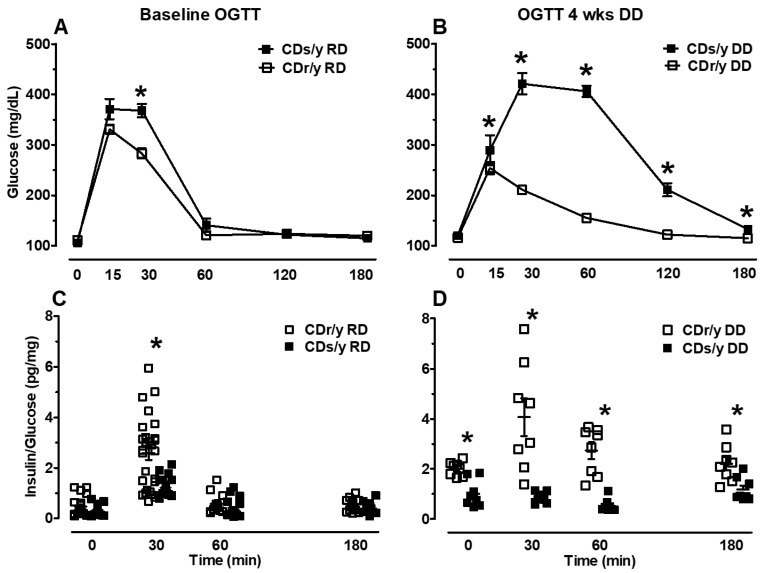
The results of an oral glucose tolerance test (OGTT): Glucose levels in CDs/y and CDr/y (panel **A**) at baseline while fed RD (*n* = 9 and *n* = 10, respectively), and (panel **B**) after 4 weeks of DD (*n* = 14 and *n* = 14, respectively)–data are shown as average ± SEM; insulin/glucose ratio in CDs/y and CDr/y (panel **C**) at baseline while fed RD (*n* = 13–27 and *n* = 16–25, respectively) and (panel **D**) after 4 weeks of DD (*n* = 8 and *n* = 8, respectively)–data are shown as scattergrams. Between group (CDs/y vs. CDr/y) analysis at each time point was by two-tailed Student’s *t*-test, * *p* < 0.01.

**Figure 5 ijms-24-01355-f005:**
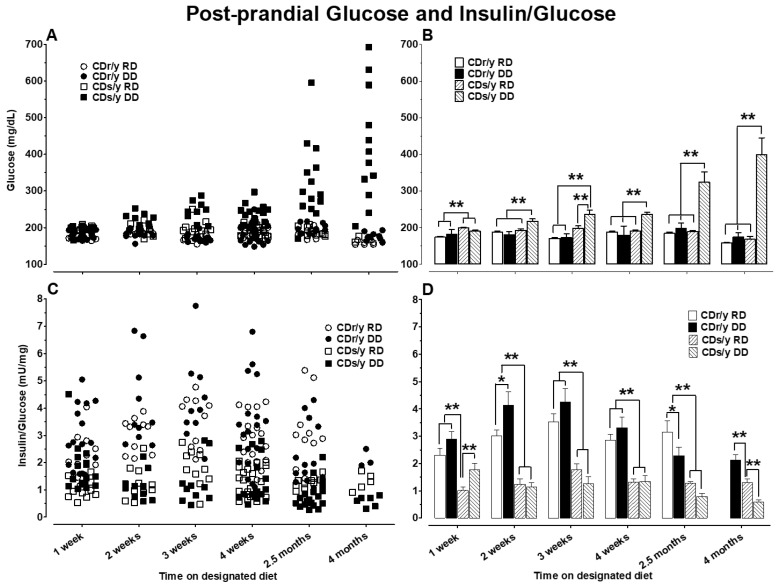
Post-prandial glucose levels and insulin/glucose ratio, shown as scattergram in panels (**A**) and (**C**), respectively and as vertical bars with SEM in panels (**B**) and (**D**), respectively, in CDs/y and CDr/y fed RD or DD after 1–4 weeks (*n* = 10–19 in each group), 2.5 months (*n* = 10–19 in each group) and 4 months (*n* = 3–6 in each group) from initiation of dietary intervention. Between group analysis was by ANOVA and post-hoc LSD testing, * *p* < 0.05, ** *p* < 0.01.

**Figure 6 ijms-24-01355-f006:**
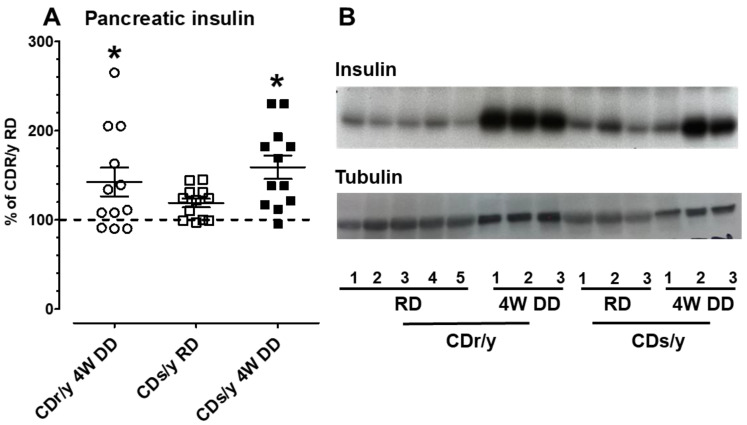
Panel (**A**)–pancreatic insulin content by Western blot and densitometry in CDr/y provided RD (*n* = 10), in CDr/y provided DD over 4 weeks (open circles, *n* = 12), in CDs/y provided RD (open squares, *n* = 12) and in CDs provided DD for 4 weeks (closed squares, *n* = 12). The dotted horizontal line represents the average of ten animals, assayed as five duplicates and serving as reference in each gel to standardize for between gel-variations in densitometry readings. * *p* < 0.01 compared to CDr/y RD by one way ANOVA and post-hoc LSD testing. Panel (**B**)-representative Western blot of insulin in each group and tubulin which served as the internal protein control. For CDr/y RD, we used ten animals pooled in five duplicates (labeled 1–5); for CDr/y DD, CDs/y RD, and CDs/y DD, we loaded on each gel protein extracts from three animals in each group (labeled 1–3).

**Figure 7 ijms-24-01355-f007:**
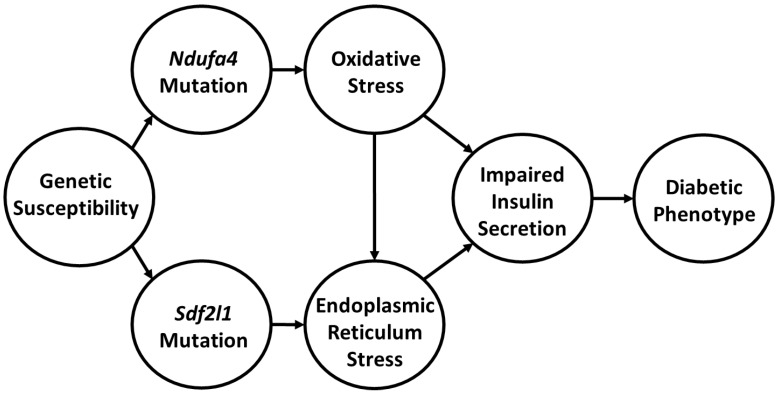
Proposed multi-hit mechanism for the development of the diabetic phenotype in CDs/y.

## Data Availability

All data are available in the records of the Laboratory for Molecular Medicine at the Barzilai University Medical Center in Ashkelon, Israel.

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
