# Peer review of "Dysregulated UPR and ER Stress Related to a Mutation in the Sdf2l1 Gene Are Involved in the Pathophysiology of Diet-Induced Diabetes in the Cohen Diabetic Rat"

_ijms, 2023, doi:10.3390/ijms24021355_

Round 1
Reviewer 1 Report
Title: Dysregulated UPR and ER stress related to a mutation in the Sdf2l1 gene are involved in the pathophysiology of diet-induced diabetes in the Cohen Diabetic rat
Authors: Chana Yagil , Ronen Varadi-Levi , Yoram Yagil *
In this study authors evaluated the effects of deletion of SDF2L1 gene expression in rat models of type 2 diabetes mellitus. They observed that this deletion in the diabetic models induced dysregulation in pathways that are involved in protein folding and ER stress, concluding that this may contribute to the pathophysiology of diabetes in their animal model.
This investigation contributes to the evaluation of the pathophysiology of diabetes and highlights deletion that may contribute to the disease of diabetes of type 2. The paper is suitable for the special issue “Cellular and Molecular Mechanisms of Cardiovascular and Metabolic Diseases”, however some criticisms arise, and the manuscript should be revised before publication.
Major and minor comments
In the introduction the link between SDF2L1 and the unfolded protein response should be better described, indicating which pathways and proteins are involved. The paragraphs of the discussion, in which there is an explanation of the UPR pathways, could be moved to the introduction. This would allow to shorten the long discussion.
The legend of figure 1 does not describe the image. It must be improved by explaining protein cascades and presenting a list of abbreviations.
Fig.2 Did authors quantify protein expression? Mw of proteins should be provided in the picture.
Fig. 3 A representative blot of each protein should be added, and statistical analysis should be consistent with the graphical representation.
The text requires some corrections to punctuation.
Reviewer 2 Report
Major revision:
Interesting paper looking at the genetic component of the UPR in rats.
Figure 1: need to expand to include BiP, pGSK, JNK, and GAPDH
Figure 2: Should look at brain as well
Figure 3: should add marker for arm 3
Figure 4: Should add weight measurements.
Figure 5: data is sufficient
Figure 6: should measure growth hormone levels
Figure 7: figure is sufficient
The paper would greatly benefit from expansion regarding how the pathway can be targeted PMID: 28077004 and what effects this can have on behavior PMID: 25540611
If the above are addressed and references included, paper could be of interest.
Reviewer 3 Report
The present manuscript is very interesting and well written and designed however I have some minor points:
1- The types of the tissues used in the study not mentioned in the method section
2- Figure 3: All the comparisons are relative to the CDr/y RD, while no comparison between the two strains on DD (CDs vs. CDr
3- Figure 5: the scattergrams (A and C) are confusing and redundant
4- The authors should outlined the limitations of their study
Author Response
Please see the attachement. Thak you

Round 2
Reviewer 1 Report
The authors addressed reviewer’s comment and have made substantial modifications strengthening the quality of the manuscript. I have no further comments at this time.